# Influence of the Carrier Gas Flow in the CVD Synthesis of 2-Dimensional MoS_2_ Based on the Spin-Coating of Liquid Molybdenum Precursors

**DOI:** 10.3390/nano14211749

**Published:** 2024-10-31

**Authors:** Fiorenza Esposito, Matteo Bosi, Giovanni Attolini, Francesca Rossi, Roberto Fornari, Filippo Fabbri, Luca Seravalli

**Affiliations:** 1Institute of Materials for Electronics and Magnetism—National Research Council (IMEM-CNR), Parco Area delle Scienze 37/A, 43124 Parma, Italy; fiorenza.esposito@imem.cnr.it (F.E.); matteo.bosi@imem.cnr.it (M.B.); giovanni.attolini@imem.cnr.it (G.A.); francesca.rossi@imem.cnr.it (F.R.); roberto.fornari1@unipr.it (R.F.); 2Department of Chemical Science, Life, and Environmental Sustainability, University of Parma, 43124 Parma, Italy; 3Department of Mathematical, Physical and Computer Sciences, University of Parma, 43124 Parma, Italy; 4NEST, Istituto Nanoscienze—CNR, Scuola Normale Superiore, Piazza San Silvestro 12, 56127 Pisa, Italy

**Keywords:** molybdenum disulfide, liquid molybdenum precursors, chemical vapor deposition, carrier gas, Raman spectroscopy, photoluminescence spectroscopy

## Abstract

Atomically thin molybdenum disulfide (MoS_2_) is a two-dimensional semiconductor with versatile applications. The recent adoption of liquid molybdenum precursors in chemical vapor deposition has contributed significantly to the reproducible wafer-scale synthesis of MoS_2_ monolayer and few-layer films. In this work, we study the effects of the carrier gas flow rate on the properties of two-dimensional molybdenum disulfide grown by liquid-precursor-intermediate chemical vapor deposition on SiO_2_/Si substrates. We characterized the samples using Optical Microscopy, Scanning Electron Microscopy, Raman spectroscopy, and Photoluminescence spectroscopy. We analyzed samples grown with different nitrogen carrier flows, ranging from 150 to 300 sccm, and discussed the effect of carrier gas flows on their properties. We found a correlation between MoS_2_ flake lateral size, shape, and number of layers, and we present a qualitative growth model based on changes in sulfur provision caused by different carrier flows. We show how the use of liquid precursors can allow for the synthesis of homogeneous, single-layer flakes up to 100 µm in lateral size by optimizing the gas flow rate. These results are essential for gaining a deeper understanding of the growth process of MoS_2_.

## 1. Introduction

The synthesis of a 2D material is a crucial step because different preparation processes can determine its fundamental material properties [1,2,3,4,5,6]. Among the various techniques to obtain atomically thin structures from a bottom-up approach, Chemical Vapor Deposition (CVD) has established itself as the most successful, with the recent demonstration that CVD based on metalorganic sources (MOCVD) is the most promising option for wafer-scale synthesis [7,8,9,10,11]. Among 2D materials, molybdenum disulfide (MoS_2_) has attracted significant attention due to its unique optical and electrical properties, including a direct bandgap, strong light–matter interaction, and high quantum yield [12]. Various applications for this 2D material in the photonics field have been proposed, such as photodetectors, 2D LEDs, single photon sources, and components of nanophotonic circuits. We refer the reader to several excellent reviews dedicated to this topic [13,14,15].

The CVD growth of transition metal dichalcogenides is highly sensitive to reaction parameters such as precursor concentration and temperature. Therefore, any modification or uncontrolled fluctuation can significantly influence the reaction kinetics and material quality, leading to poor reproducibility in the synthesis [16]. Liquid molybdenum precursors have been introduced as a viable option to grow 2D MoS_2_ by CVD with large flake sizes, proving to be much more reliable than previously used powder precursors, which, despite initial promising results and the simplicity of the process, suffer from serious reproducibility issues [17,18,19,20,21]. This method relies on the sulfurization of a solution containing molybdenum precursors that is deposited by spin-coating on the substrate which is then loaded into the CVD reactor [22,23]. The typical solution is composed of three different components: (i) the water-soluble Mo precursor, (ii) the growth promoter, and (iii) the density gradient medium, which improves adhesion to the substrate surface during spinning. The first two components are mixed in ratios ranging from 1:1 to 8:1 [24,25].

The use of a molybdenum liquid precursor represents a significant modification to the paradigm of the CVD synthesis of two-dimensional materials. With solid precursors, both molybdenum and sulfur are provided to the growth substrate in vapor form, while in the liquid-precursor-intermediate CVD process (LPI-CVD), the solution containing the molybdenum precursor is already present on the growth substrate and undergoes oxidation and solidification during the temperature ramp [25]. This process is somewhat similar to the early work on the sulfurization of the molybdenum layer deposited on the growth substrates, where the primary drawback was that the MoS_2_ domains were limited by the initial size of the molybdenum grains in the deposited layer [26]. Although highly efficient outcomes in terms of reproducibility of the process have been obtained with Mo liquid precursors, the dynamic of the growth process has been somewhat overlooked, despite the complexity of the physical and chemical mechanisms involved. For instance, the issue of six-fold defective domains has been noted in the case of WS_2_ obtained by LPI-CVD [27]. Similarly, the synthesis of highly tensile-strained MoS_2_ monolayers has been demonstrated within a specific temperature range using liquid precursors [28]. Only recently have the different results in terms of material quality using different density gradient media been highlighted [29]. Likewise, Senkic et al. [30] reported variations in the morphology of MoS_2_ structures grown with different S:Mo ratios using liquid precursors. Nevertheless, the LPI-CVD has proven to be a successful approach for growing more complex two-dimensional structures, such as patterned MoSe_2_ [31], highly oriented MoS_2_ mono and bilayer structures [17], metal-doped monolayers [32], and van der Waals heterostructures [33]. Since the sulfurization process of liquid precursors spun on the substrate still relies on sulfur vapors being transported to the growth substrate, the flow rate of the inert carrier gas plays a crucial role as it affects both vapor concentration and sulfur precursor transportation. While several studies have focused on the effect of carrier gas in CVD growth using Mo solid precursors [33,34,35], very few have investigated the effect of carrier gas flow when using liquid precursors.

In this work, we demonstrate that the flow rate of the carrier gas is a critical parameter for the CVD growth of MoS_2_ structures, even when using Mo liquid precursors. Unlike the case of the Mo solid precursor, this parameter drastically affects the number of layers and flake sizes but has a lesser impact on the flake shape. At the lowest flow rate, structures with an inhomogeneous number of layers are obtained, featuring a bulk-like center (often pyramidal in shape) and bilayer structures along certain crystallographic directions. At the highest flow rate, the synthesis yields a 100 μm large triangular monolayer with concave edges. Notably, increasing the flow rate decreases the number of layers, reaching the monolayer limit.

## 2. Materials and Methods

The solution deposited on the substrate consists of a mixture of (i) 1 mL of ammonium heptamolybdate tetrahydrate (NH_4_)_6_Mo_7_O_24_∙_4_H_2_O (AHT, Sigma-Aldrich (St. Louis, MO, USA), 99.98%) 0.030 mol/L, (ii) 4 mL of NaOH 0.060 mol/L, which acts as a growth promoter, and (iii) 0.5 mL of OptiPrep^TM^, a density gradient medium containing 60% iodixanol and 40% water. In total, 10 µL of this mixture is then used for spin-coating on a 1 × 1 cm^2^ SiO_2_/Si substrate (3000 rpm, 30 s) which has been previously treated with O_2_ plasma (25 W, 3 min) to improve wettability. The substrate is placed 20 cm away from an alumina boat containing 275 mg of a sulfur solid precursor (Sigma-Aldrich, 99.98%). The growth process is carried out at a temperature of 820 °C (heating rate of 40 °C/min), providing optimal growth conditions, while the low zone containing sulfur powders is independently heated to 180 °C (heating rate of 60 °C/min). The nitrogen carrier gas flow rate, selected for its inert nature and low cost, is kept constant within the range of 150 to 300 sccm during the growth process. Further details on the CVD growth system can be found in [34,35].

The morphology of the MoS_2_ flakes is analyzed by scanning electron microscopy (SEM) using a Zeiss Auriga Compact system equipped with a GEMINI Field-Emission column. A statistical analysis of the lateral dimensions of the flakes is performed using SEM images. For each sample, approximately one hundred flakes distributed across different areas of the substrate are considered.

Raman and photoluminescence measurements are carried out with a Renishaw InVia system, equipped with a confocal microscope, a 532 nm excitation laser, and an 1800 line/mm grating (spectral resolution < 2 cm^−1^). All the analyses are performed with a 100X objective (NA = 0.85), excitation laser power 500 μW, and an acquisition time of 4 s. The Raman mapping is carried out with a pixel size between 800 nm and 1 μm.

## 3. Results

We observed that the density and the size of the flakes can be reproducibly controlled by adjusting the carrier gas flow rate. In Figure 1, optical microscopy (OM) and scanning electron microscopy (SEM) images are shown for samples grown under different carrier flows, ranging from 150 to 300 sccm. Lower and higher gas flow rates were also explored, resulting in very small flakes at values below 150 sccm and low-quality 3D structures at higher values.

The concavity of the flakes and thus their shape is directly influenced by the Mo:S ratio. A perfect triangular shape is typically reported in studies using solid precursors, achieved by increasing the sulfur content and growth temperature, whereas greater concavity (multi-apex triangles) is observed when the growth occurs under Mo excess [36]. Statistical measurements of flake concavity show values ranging from 7.5° to 10.2° for carrier flows between 150 and 250 sccm (Figure 2). A higher concavity, reaching 12.5°, was found for the sample with a flow rate of 300 sccm. This analysis suggests that the growth behavior using liquid precursors differs significantly from that observed with solid precursors (such as MoO_3_ powders), where carrier gas flow, gas velocity, MoO_3_ and S transport, and gas phase pre-reactions greatly influence the growth outcome [37,38,39]. As shown in Figure 2e, flake concavity remains almost constant for flows up to 250 sccm but increases significantly at 300 sccm.

Figure 3 presents the statistical analysis based on SEM images and the average lateral size of the flakes. By fitting the histograms in Figure 3a–d with Gaussian functions, the average flake sizes were found to be 5.6 ± 0.7 µm, 19.8 ± 2.9 µm, 12.0 ± 3.5 µm, and 76.7 ± 9.1 µm for flows rates of 150 sccm, 200 sccm, 250 sccm, and 300 sccm, respectively. As shown in Figure 3e, data indicate a nonlinear increase in flake size as the flow rate increases from 150 to 300 sccm.

In Figure 4, we present maps of the separation between Raman modes (Δk), a well-established method for determining the number of MoS_2_ layers. Monolayer structures exhibit Δk < 19 cm^−1^, while bulk structures show Δk > 24 cm^−1^ [40,41]. Optical reference images of the analyzed flakes, as well as representative Raman spectra, are shown in Appendix A.

The flakes obtained with a flow of 150 sccm consist of a bulk-like central region and three bilayer lateral structures, as evidenced by the separation of the Raman modes: 26 cm^−1^ in the central region (bulk-like) and 20 cm^−1^ in the lateral areas. The flakes grown with a carrier gas flow rate of 200 sccm and 250 sccm exhibit a homogeneous spatial distribution of Δk. Specifically, Δk is equal to 23 cm^−1^ and 21 cm^−1^ in the case of the 200 sccm and 250 sccm specimens, respectively. These values indicate the presence of few-layer structures. The flakes obtained with a flow of 300 sccm are large, corner-shaped triangular monolayers with Δk = 19 cm^−1^. As recently reported [28], the separation of the Raman modes at the corner of the triangular structure is influenced by tensile strain.

Photoluminescence (PL) characterization was performed to analyze the effect of varying carrier gas flow rates on the optical properties [37,38,42].

Figure 5 presents the PL spectra of the triangular structures obtained at different carrier gas flows. For the sample grown at 150 sccm, we report the PL spectrum obtained from the lateral part of the flakes, as the central bulk-like area does not exhibit any light emission. The PL spectra reveal two distinct light emissions, corresponding to the A and B excitonic recombinations of MoS_2_ [43,44,45,46]. In the samples grown at 150 sccm and 300 sccm, the A exciton peaks at 1.83 eV, whereas the samples obtained at 200 sccm and 250 sccm show a redshift to 1.78 eV. The B exciton consistently peaks at 1.95 eV. Notably, the intensity ratio of A to B varies significantly: it is 2.1 for the 200 sccm and 250 sccm samples, increasing to 8 and 10.2 for the 150 sccm and 300 sccm samples, respectively. The PL intensity ratios and peak positions (Figure 5) confirm the findings from the Raman spectroscopic mapping (Figure 4), indicating that the number of layers varies with the carrier gas flow during the growth process. As the distance between the two Raman peaks decreases—an indication of there being few layers—the integrated area of the exciton A peak in the PL spectrum increases [45]. Both these features suggest the presence of an unstrained MoS_2_ monolayer.

## 4. Discussion

Most of the literature on MoS_2_ synthesis focuses on solid precursors, which is not directly applicable to our case. Liu et al. [47] demonstrated that increasing the carrier gas flow rate enhances the rate of precursor transport along the growth tube. Furthermore, Cao et al. [48] revealed that the carrier gas flow rate strongly influences the structural and morphological properties of MoS_2_ flakes during CVD growth, which subsequently affects their optical properties. In particular, they observed dendritic structures at higher carrier flow rates, while lower carrier flows resulted in triangular shapes. Additionally, it was shown [49] that the concentration of sulfur vapor determines the size and shape of CVD-grown MoS_2_ flakes, with the flake shape changing from triangle to hexagon and back to triangle as the sulfur concentration decreased.

The growth process with liquid precursors differs markedly from the typical synthesis of MoS_2_ flakes using solid sources in terms of species transport and the reactions involved. Unlike solid Mo powders, the molybdenum available for MoS_2_ synthesis with liquid precursors is controlled in a highly reproducible manner by applying a solution spun on the substrate. This method avoids the transport processes of MoO_3_ powder in the gas phase, the heterogeneous gas phase reactions between Mo and S, and the uncontrolled MoO_x_ sub-oxidation processes that usually occur in the gas phase prior to the nucleation on the substrate. All these processes, which occur with solid sources, create a complex relationship between the carrier flow and the concentration of the precursors in both the gas phase and on the substrate surface, potentially leading to reproducibility issues [50,51]. By eliminating the role of gas phase reactions, we hypothesize that all the sulfur evaporating from the source will reach the substrate for the synthesis reactions: no sulfur is lost due to heterogeneous gas phase reactions, as in the case with solid precursors, and no sulfur deposition is experimentally observed in the hot zone of the furnace, between the sulfur boat and the substrate. We highlight that the temperature profile along the tube was measured prior to deposition, and no significant gradients were found in the zone relevant to the deposition. Moreover, uniform gas mixing between sulfur and nitrogen is expected due to the gas flow rate, the high temperature, and the significant distance between the sulfur source and the growth substrate.

The quantity of sulfur available for growth is defined by the solid source temperature and by its dilution in the transport carrier gas. The local concentration of sulfur in the gas stream depends on the flow dynamics of the system, which are influenced by the diameter of the reaction tube, the carrier gas flow rate, the evaporated sulfur, and the temperature profile of the system. The distance between the sulfur source and the substrate may also influence how sulfur is distributed in the gas stream and its local concentration over the liquid precursor. The sulfur available for growth is determined by the number of atoms reaching the substrate surface from the gas phase, primarily defined by the flow dynamics. The sulfur vapor pressure is set by the temperature of the sulfur powder: being in an open tube, the system is in a dynamic equilibrium, and we can assume that the sulfur vapor will never reach its equilibrium pressure. Consequently, the quantity of sulfur evaporating is constant at a given temperature, but its dilution and partial pressure in the gas phase depend on the total amount of the carrier gas: a higher gas flow in standard cubic centimeters per minute results in greater sulfur dilution in the growth tube because the same quantity of sulfur is evaporated and diluted in a larger volume of gas. Thus, we can express the sulfur partial pressure P.P.(S) as inversely proportional to the total flow F_tot_:$$P.P.\;\left(S\right)\;\sim \frac{1}{F_{tot}}$$

Under our experimental conditions, the flow is laminar (the Reynolds number with a N_2_ flow of 300 sccm is approximately 14). The carrier gas flow affects the thickness of the boundary layer between the nutrient gas phase and the substrate surface, influencing sulfur diffusion from the nutrient gas phase to the surface and the diffusion of the reaction byproducts from the surface to the gas phase. The velocity of sulfur diffusion towards the surface is inversely proportional to the boundary layer thickness, which depends on $\frac{1}{\sqrt{F_{tot}}}$ [52,53]. It has also been observed that, in the sulfurization of a molybdenum thin layer deposited by sputtering, the rate-limiting step is the diffusion of sulfur to the surface, highlighting the pivotal role of sulfur diffusion in a system similar to ours and supporting the relevance of the boundary layer model [54,55].

Combining these two dependencies, we can estimate that the sulfur concentration at the substrate surface is proportional to the following:$$surface\;S\;concentration\;\sim \;P.P.\;\left(S\right)\times \frac{1}{boundary\;layer\;thickness}\;\sim \;\frac{1}{F\;}\times \sqrt{F}\;\sim \frac{1}{\sqrt{F}}$$

We therefore expect a higher sulfur concentration at the substrate surface at low carrier gas flows, and conversely, a lower sulfur surface concentration at higher flows.

In the model proposed here, we make several assumptions derived from previous works using solid molybdenum precursors, particularly regarding the effect of sulfur supply on flake growth. It is important to emphasize that the change in flow rate affects the growth of flakes using solid and liquid precursors in entirely different ways. The transport of molybdenum vapor from solid sources and reactions between MoO_3_ and S occurring in the gas phase prior to the synthesis on the flakes do not occur in our system [10,56,57]. By using liquid molybdenum precursors deposited by spin-coating on the growth substrate, the quantity of the molybdenum precursor can be considered constant, independent of other growth parameters. Following the seminal work of Kim et al. [25], during the temperature ramp, the AHT reacts with the residual oxygen in the chamber, creating a mixture of MoO_3_ and Na_2_MoO_4_.

The orientation of the MoS_2_ structures may also be affected, as changes in the S:Mo ratio can lead to the synthesis of vertically aligned MoS_2_ microscale structures [58]. Additionally, variations in the S:Mo can alter the CVD process, favoring a more layer-by-layer growth mechanism and reducing the desired two-dimensional in-plane enlargement [59].

In Figure 6, we illustrate the effect of the different sulfur provisions on the number of layers and the shape of the obtained MoS_2_ structures, based on total carrier flow. Considering the flow dynamic model described above, lower carrier flows result in higher sulfur supply (left panel), leading to MoS_2_ structures composed of a central layer-by-layer pyramidal structure with lateral bilayer features [60,61]. A medium sulfur supply (central panel) results in the growth of few-layer structures. Finally, increasing the flow decreases the sulfur provision, resulting in monolayer large-area structures.

The change in sulfur supply transitions the growth regime from kinetic driven (at low flow/high sulfur concentration at the surface) to thermodynamically driven (at higher flow/lower sulfur concentration), which may explain the observed changes in flake shape: at lower flows, the higher concentration of sulfur adatoms arriving at the substrate inhibits the enlargement of existing flakes, as it becomes more favorable to nucleate a new MoS_2_ seed rather than enlarging an existing one, thus explaining the formation of pyramid structures and the aggregation of flakes. As the carrier flow increases, the supersaturation of sulfur at the surface decreases and the growth becomes more thermodynamically driven; enlarging existing seeds becomes favorable compared to nucleating new ones, resulting in larger and more regular flakes.

By eliminating the gas phase heterogeneous reactions present when using solid precursors, which depend on many parameters (temperature, distance between the Mo and S powders, position of the substrate, etc.), the growth process with liquid precursors is more controlled and reliable. This may enhance our understanding of MoS_2_ growth mechanisms. It should also be noted that, for this reason, results obtained with solid precursors cannot be directly explained by this model.

We also discuss the modifications in the shape of the MoS_2_ structures obtained with different sulfur provisions. It is generally observed that concavity increases, and multi-apex triangles are formed if the growth occurs in molybdenum excess [41]. In the case of a lower provision of sulfur, the shape indicates that the CVD process is carried out in a molybdenum-rich environment, as the structures exhibit undergrowth along the sulfur in zig-zag directions [62]. Increasing the sulfur supply leads to a sulfur-rich environment, resulting in concave triangular structures [63].

The increased concavity in the specimen obtained with a 300 sccm flow rate can be related to a slight decrease in the S:Mo ratio under these conditions, which also induces an increase in the lateral size. Therefore, at higher flows, the increased concavity of flakes suggests that the growth occurs with a slightly lower sulfur supply, corroborating the proposed model regarding sulfur provision. This finding contrasts with what is generally observed with solid precursors, where a higher gas flow rate is reported to increase the provision of sulfur atoms, leading to flakes with reduced concavity and a more regular triangular shape. This discrepancy, as mentioned in the previous section, may be attributed to the very different growth dynamics in the two systems and the absence of gas phase pre-reactions between MoO_3_ and S when using solid precursors.

The use of liquid precursors simplifies the growth mechanism of MoS_2_ flakes by separating the sulfur supply, which is controlled solely by flow dynamics, from the surface reactions governed by kinetic and thermodynamic processes. This approach also clarifies the influence of sulfur supply on the shape and size of MoS_2_ flakes, as it depends only on the total carrier flow, which is proportional to $\frac{1}{\sqrt{F}}$. Our model demonstrates that the S:Mo ratio in the case of liquid precursor CVD simultaneously affects both the number of layers and the size of the MoS_2_ structures. At a high sulfur supply (carrier flow = 150 sccm), we obtain small structures characterized by a bulk central region with lateral overgrowth. Conversely, as the sulfur supply decreases, few-layer structures form. When the sulfur reaching the substrate is lower (carrier flow = 300 sccm), monolayer MoS_2_ flakes are produced with an average lateral size that is 3.5 times larger than those grown under higher sulfur supply conditions (carrier flow = 200–250 sccm). While our method allows for the growth of large flakes, obtaining a uniform continuous film with our CVD system configuration remains a challenge. To date, this has only been achieved using vertical substrate positioning [64].

To summarize the progress in 2D MoS_2_ growth using different approaches, we provide Table 1, where we highlight some relevant parameters from representative articles. Due to the vast number of scientific publications on this material, this table should be considered only as a guideline for standard values commonly found in the literature.

## 5. Conclusions

We investigated the influence of the carrier gas flow rate on the synthesis of two-dimensional MoS_2_ structures using Mo liquid precursors by analyzing the properties of samples grown under identical conditions with varying carrier flows. Our results demonstrate that adjusting the carrier gas flow allows control over the nucleation density, morphology, lateral size, and the number of layers in the MoS_2_ flakes. At the lowest flow rate, the resulting structures exhibit an inhomogeneous number of layers, featuring a bulk-like center (predominantly pyramidal) with bilayer structures extending along the Mo zig-zag crystallographic directions. At the highest flow rate, the synthesis yields 100 μm large triangular monolayers with concave edges. These findings are in stark contrast to the results commonly reported for MoS_2_ 2D structures grown using solid Mo precursors.

We propose a model to explain the effect of different carrier flows on the properties of the flakes, based on changes in sulfur provision: low flows create sulfur-rich conditions, leading to small, layer-by-layer pyramidal structures with lateral bilayers, whereas high flows reduce the S:Mo ratio, resulting in larger, monolayer flakes with greater concavity. This discussion highlights the key differences in the growth process between solid and liquid Mo precursors, emphasizing that carrier gas flow is a critical parameter in the CVD growth of MoS_2_ structures. Our study shows that, by optimizing the CVD growth parameters, it is possible to reproducibly achieve monolayer flakes with an average lateral size of about 100 µm.

## Figures and Tables

**Figure 1 nanomaterials-14-01749-f001:**
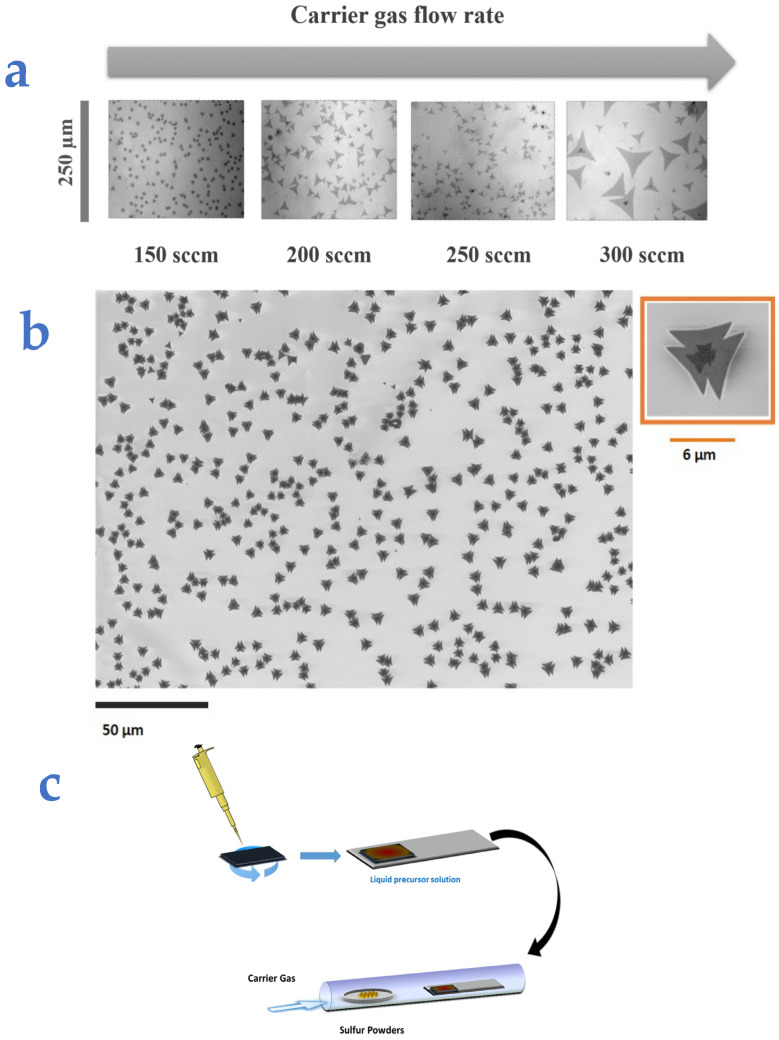
(**a**) Optical microscopy images of flakes at different nitrogen flows. (**b**) SEM image of the sample with at a flow rate of 150 sccm. (**c**) Schematic of the synthesis procedure.

**Figure 2 nanomaterials-14-01749-f002:**
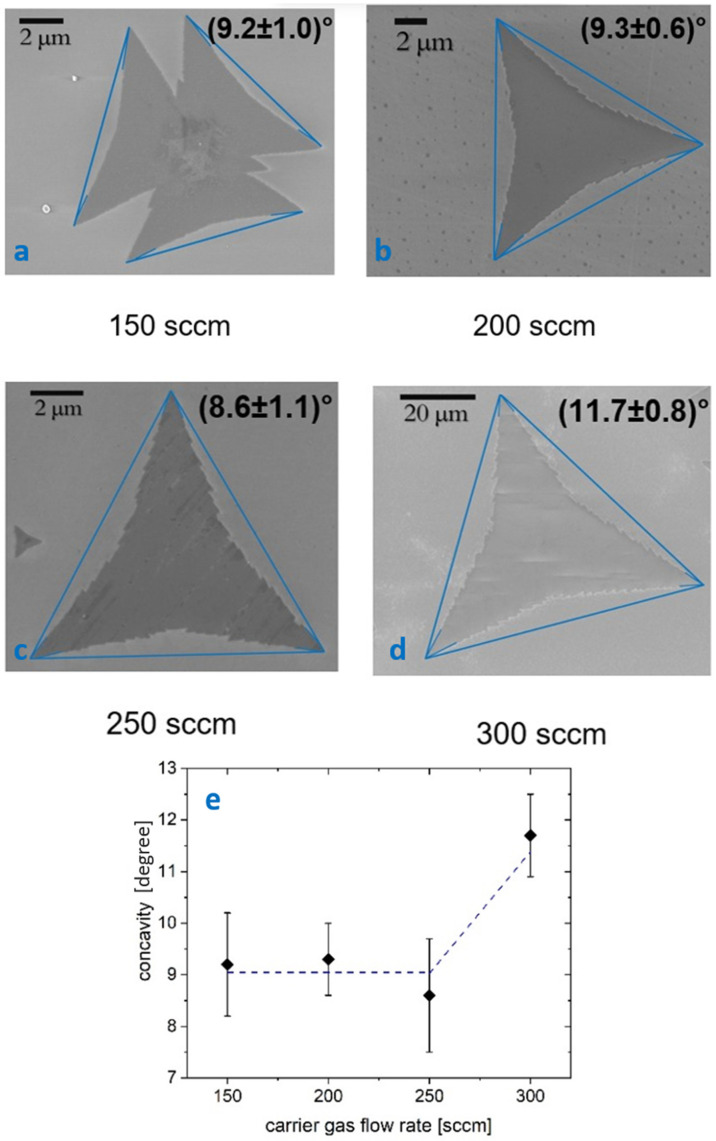
(**a**–**d**) SEM images and flake concavity for flow rates of 150, 200, 250, 300 sccm, respectively. (**e**) Concavity values plotted as a function of flow rate, with the dashed line as a guide for the eye.

**Figure 3 nanomaterials-14-01749-f003:**
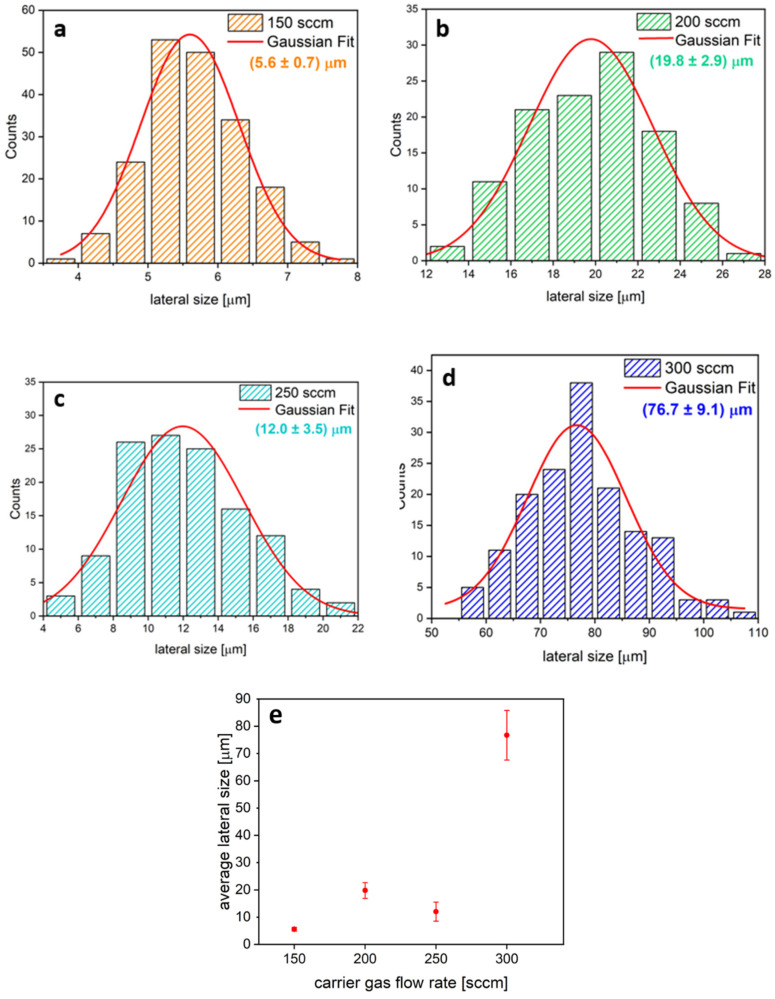
(**a**–**d**) Statistical analysis of the lateral dimensions of MoS_2_ flakes at different flow rates (150, 200, 250, 300 sccm). (**e**) Average lateral size of flakes as a function of the carrier flow rate.

**Figure 4 nanomaterials-14-01749-f004:**
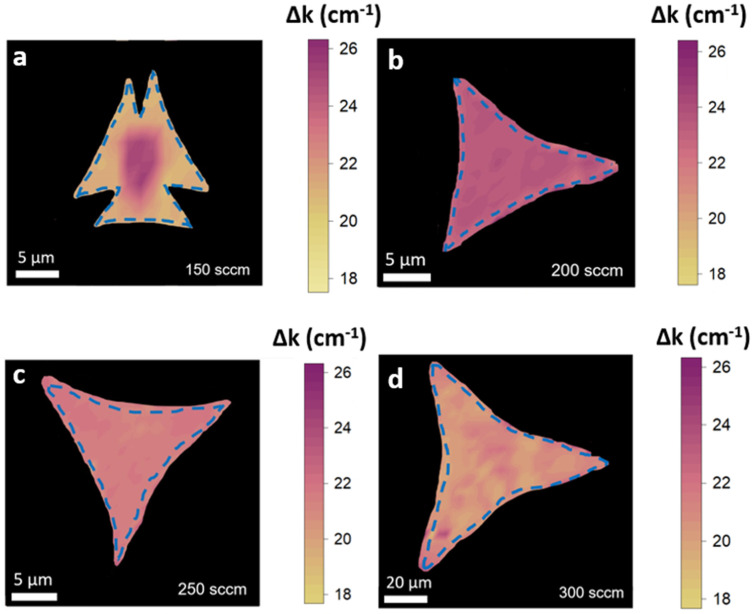
(**a**–**d**) Raman mode separation maps of MoS_2_ at flow rates 150-200-250-300 sccm. Yellow areas indicate monolayer and bilayer MoS_2_, while pink and purple areas indicate few-layer MoS_2_.

**Figure 5 nanomaterials-14-01749-f005:**
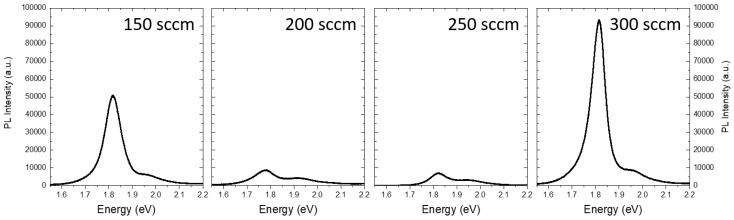
PL spectra for samples at flow rates of 150-200-250-300 sccm.

**Figure 6 nanomaterials-14-01749-f006:**
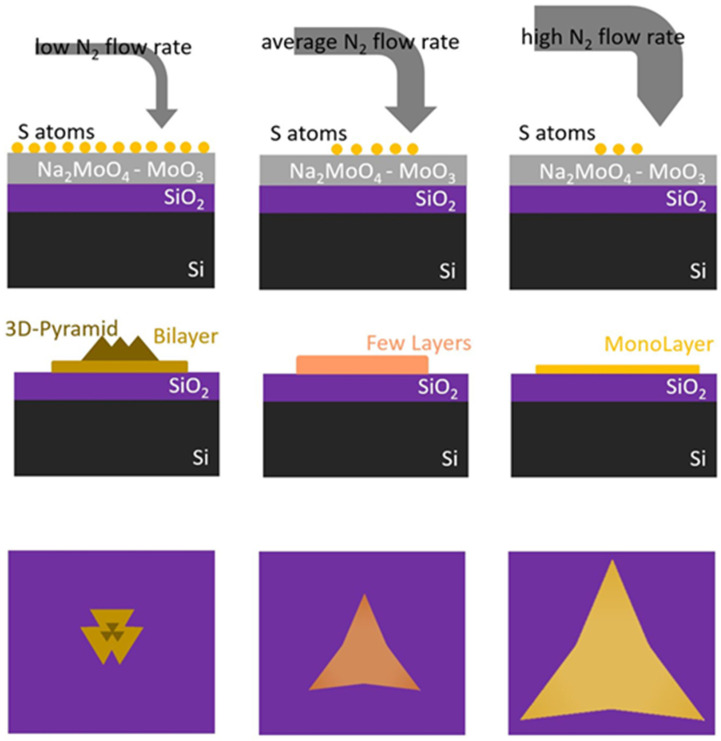
A sketch illustrating the influence of the carrier gas flow rate on sulfur supply, as well as on the shape, size, and number of layers of the flakes, with side and top views of the MoS_2_ structures.

**Table 1 nanomaterials-14-01749-t001:** Overview of recent studies on 2D-MoS_2_ growth, summarizing key parameters from selected publications to provide standard reference values for different growth approaches.

Flake Size (µm)	Mo Precursor	Temperature	Substrate	No. of Layers	Applications	Ref.
continuous	MoO_3_ thin film	1000 °C	c-sapphire	Trilayers	Electronics (FET)	[65]
10–50	MoO_3_ powders	750 °C	SiO_2_/Si	Mono and Bilayers	Photonics	[34]
10–100	MoO_3_ powders	700–720 °C	SiO_2_/Si	Monolayers	n.a.	[36]
10–20	Liquid	850 °C	c-sapphire	Monolayers	Electronics (FET)	[17]
100–250	Liquid	725 °C	SiO_2_/Si	Mono and Bilayers	Electronics (FET)	[19]
100–200	Liquid	820 °C	SiO_2_/Si	Monolayers	n.a.	This work

## Data Availability

The data are available upon reasonable request from the corresponding author.

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
