# Peer review of "Influence of the Carrier Gas Flow in the CVD Synthesis of 2-Dimensional MoS2 Based on the Spin-Coating of Liquid Molybdenum Precursors"

_nanomaterials, 2024, doi:10.3390/nano14211749_

Round 1
Reviewer 1 Report
Comments and Suggestions for Authors
MoS2 is a widely studied 2D material. In this work, the authors study the effect of the gas flow on the growth of MoS2. It is more like a lab report rather than a systematic and insightful scientific research. For example, the other factors that might influence the growth such as temperature are not considered. With a single variant, the obtained conclusion is not really useful for practical synthesis. Also, the study only includes 4 data points for different gas flow rates and it makes little sense for the scientific community. I am not able to recommend it for publication.
Author Response
We sincerely thank the reviewers for their thoughtful comments and the time they dedicated to reviewing our manuscript. We also appreciate their recognition of the clarity in presenting the state of the art on the subject, as well as our results. Ensuring that our writing is clear and easy to understand has always been a prerequisite for us when preparing our articles.
Below, we address each of the reviewers' comments in detail, providing additional literature references that support and validate our viewpoint.
Reviewer 1
MoS2 is a widely studied 2D material. In this work, the authors study the effect of the gas flow on the growth of MoS2. It is more like a lab report rather than a systematic and insightful scientific research. For example, the other factors that might influence the growth such as temperature are not considered. With a single variant, the obtained conclusion is not really useful for practical synthesis. Also, the study only includes 4 data points for different gas flow rates and it makes little sense for the scientific community. I am not able to recommend it for publication.
Thank you for your valuable feedback. We would like to clarify that the effect of temperature on MoS2 growth has already been studied by us in a published paper (Ref. 25) and, moreover, it has been explored in depth in many articles, thus it was not the focus of our work.
Seravalli, L., et al. (2023) "Built-in tensile strain dependence on the lateral size of monolayer MoS2 synthesized by liquid precursor chemical vapor deposition." Nanoscale 15.35: 14669-14678.
Senkić, Ana, et al. (2023) "Effects of CVD growth parameters on global and local optical properties of MoS2 monolayers." Materials chemistry and physics 296: 127185.
Zhu, Zusong, et al. (2019) "Influence of growth temperature on MoS2 synthesis by chemical vapor deposition." Materials Research Express 6.9: 095011.
Ardahe, Mahnoosh, Mohammad Reza Hantehzadeh, and Mahmood Ghoranneviss. (2020) "Effect of growth temperature on physical properties of MoS2 thin films synthesized by CVD." Journal of Electronic Materials 49.2: 1002-1008.
Shahzad, Rauf, TaeWan Kim, and Sang-Woo Kang. (2017) "Effects of temperature and pressure on sulfurization of molybdenum nano-sheets for MoS2 synthesis." Thin Solid Films 641: 79-86.
The main objective of this study was to investigate the specific influence of the carrier gas using a liquid precursor, a specific effect that was not studied so far. In this regard, in the introduction we present a comparison of the different roles that the carrier gas plays when using a solid precursor versus a liquid precursor, another topic on which we believe this insight provides a meaningful contribution to the field, also in consideration of the novel growth model we propose. One of the main result is the difference role of the carrier gas between the commonly used solid precursors and the liquid precursors. We strongly believe that this analysis may provide meaningful insights to researchers working on this topic. We also explain with our model why we observe this result, and the model itself may be a useful starting point for material optimization in a similar system.
We demonstrate that the effect of carrier gas flow is very relevant, and we also remark that not very often this parameter is considered in papers devoted to the growth of this material. Also, the 4 data points are relative to 4 different grown samples: this number is usually considered totally sufficient to reach meaningful conclusions, as evidenced in all these papers from high impact journals.
Ahn, G. H., Amani, M., Rasool, H., Lien, D.-H., Mastandrea, J. P., Ager III, J. W., Dubey, M., Chrzan, D. C., Minor, A. M., & Javey, A. (2017). "Strain-engineered growth of two-dimensional materials." Nature Communications, 8(1), 608.
Ko, H., Kim, H. S., Ramzan, M. S., Byeon, S., Choi, S. H., Kim, K. K., Kim, Y. H., & Kim, S. M. (2020). "Atomistic mechanisms of seeding promoter-controlled growth of molybdenum disulphide." 2D Materials, 7(1), 015013.
Li, T., Guo, W., Ma, L., Li, W., Yu, Z., Han, Z., Gao, S., Liu, L., Fan, D., Wang, Z., Yang, Y., Lin, W., Luo, Z., Chen, X., Dai, N., Tu, X., Pan, D., Yao, Y., Wang, P., … Wang, X. (2021) "Epitaxial growth of wafer-scale molybdenum disulfide semiconductor single crystals on sapphire." Nature Nanotechnology, 16(11), 1201–1207.
Senkić, Ana, et al. (2023) "Effects of CVD growth parameters on global and local optical properties of MoS2 monolayers." Materials chemistry and physics 296: 127185
Reviewer 2
This paper investigates the influence of carrier gas flow rates on the synthesis of two-dimensional MoS₂ using liquid molybdenum precursors via chemical vapor deposition (CVD). By adjusting the flow rate of the carrier gas, the study analyzes variations in the morphology, number of layers, size, and other properties of MoS₂ flakes. It proposes a growth model that links sulfur provision with carrier gas flow rate and explains how this impacts the structure of MoS₂.
- The paper primarily focuses on the impact of carrier gas flow rate on flake size and morphology but provides insufficient discussion of other important parameters such as temperature and pressure. These factors likely play a significant role in determining the quality and morphology of the flakes.
Thank you for your insightful comments. We acknowledge the importance of parameters such as temperature in influencing the growth of MoS2. However, as discussed in the reply to Ref.1 the effect of temperature has already been extensively investigated in previous studies, both from our group and from other groups worldwide. Moreover, we grow at atmospheric pressure, as in most CVD growth systems for this 2D material.
Our work is specifically aimed at highlighting the effects of the carrier gas flow rate when using a liquid precursor, an area that has received much less attention. In this context, we aimed to provide new insights by demonstrating how the carrier gas impacts the morphology and quality of the flakes, with a focus on the differences compared to the use of a solid precursor.
- Although the sulfur supply model is proposed, the authors assume that sulfur availability is solely controlled by the carrier gas flow rate. It neglects the influence of other variables such as temperature gradients and gas-phase mixing, which may lead to inaccuracies in the model.
We agree that sulfur availability can be influenced by multiple factors, such as temperature gradients and gas-phase mixing. However, in this study, we intentionally fixed these parameters across the different experiments to isolate the effect of the carrier gas flow rate. Temperature profile along the tube was measured prior to deposition and no significant gradients were found in the zone meaningful for the deposition. Moreover, in our system geometry, due to the chosen gas flow rate and the high temperature inside the tube, the distance between the sulfur source and the substrate is promoting a uniform mixing of nitrogen and sulfur vapors. In any case, the laminar flow regime of our growth system ensures reproducible results from growth to growth. Possibly, flowdynamic models could help in understanding this point but they are beyond the scope of our work. By keeping the temperature, pressure, and other relevant factors constant, we ensured that the observed variations in the results were solely due to changes in the flow rate. This approach allowed us to specifically investigate the role of the carrier gas in controlling sulfur supply and its impact on MoS2 growth and to highlight the difference to the commonly used powder precursors. We have added a sentence on this point in the manuscript in section 4 (red text).
- The study highlights the advantages of liquid molybdenum precursors over solid ones but fails to provide experimental comparisons between the two. Conducting experiments that directly compare different precursor types would strengthen the conclusions about their actual effects on MoS₂ growth.
Over the years, we have extensively studied the use of solid molybdenum precursors for MoS₂ growth, and our findings have been reported in previous publications, listed below:
Golovynskyi, Sergii, et al. (2020) "Exciton and trion in few-layer MoS2: Thickness-and temperature-dependent photoluminescence." Applied Surface Science 515: 146033.
Seravalli, L., et al. (2021) "Gold nanoparticle assisted synthesis of MoS2 monolayers by chemical vapor deposition." Nanoscale Advances 3.16: 4826-4833.
Rotunno, E., et al. (2020) "Influence of organic promoter gradient on the MoS2 growth dynamics." Nanoscale Advances 2.6: 2352-2362
Irfan, Iqra, et al. (2021) "Enhancement of Raman scattering and exciton/trion photoluminescence of monolayer and few-layer MoS2 by Ag nanoprisms and nanoparticles: shape and size effects." The Journal of Physical Chemistry C 125.7: 4119-4132.
Moreover, the advantages of using liquid precursors have been extensively studied and compared with solid precursors in the last years in many other articles (see below a representative but not exhaustive list), therefore we felt that a comparison between solid and liquid precursors would have provided no novel information to the scientific community.
Guan, H., Zhao, B., Zhao, W., & Ni, Z. (2023). "Liquid-precursor-intermediated synthesis of atomically thin transition metal dichalcogenides." Materials Horizons, 10(4), 1105–1120.
Zhang, T., Fujisawa, K., Zhang, F., Liu, M., Lucking, M. C., Gontijo, R. N., Lei, Y., Liu, H., Crust, K., Granzier-Nakajima, T., Terrones, H., Elías, A. L., & Terrones, M. (2020). "Universal In Situ Substitutional Doping of Transition Metal Dichalcogenides by Liquid-Phase Precursor-Assisted Synthesis." ACS Nano, 14(4), 4326–4335.
Seo, J., Lee, J., Baek, S., Jung, W., Oh, N. K., Son, E., & Park, H. (2021). "Liquid Precursor-Mediated Epitaxial Growth of Highly Oriented 2D van der Waals Semiconductors toward High-Performance Electronics." ACS Applied Electronic Materials, 3(12), 5528–5536.
Choi, Soo Ho, et al. (2017) "Water-assisted synthesis of molybdenum disulfide film with single organic liquid precursor." Scientific reports 7.1: 1983.
Robertson J, Blomdahl D, Islam K, Ismael T, Woody M, Failla J, Johnson M, Zhang X and Escarra M (2019) "Rapid-throughput solution-based production of wafer-scale 2D MoS2" Appl Phys Lett 114 163102
For this reason, in this paper we examined the effects of carrier gas flow when using liquid molybdenum precursors, which is less explored and gives unexpected results compared to the more common approach by solid precursors.
For the above reasons, we decided not to include experimental comparisons with solid precursors, but for a comprehensive discussion on the effects of solid precursors, we refer the reviewer to our previous works that provide detailed experimental results on MoS₂ growth using solid precursors.
- While the paper includes data from Raman spectroscopy and SEM, the interpretation of these results is not in-depth. Specifically, the discussion on strain effects observed in Raman spectra is rather superficial, without a thorough exploration of their origins or mechanisms.
In addition to Raman spectroscopy and SEM data, we also presented photoluminescence (PL) data, which is an essential characterization technique for this type of material. The PL results are in full agreement with the Raman data, further supporting our conclusions. As a matter of fact, the PL intensity ratio evidences how the number of layers changes with carrier gas flow. It should also be noted that in many articles devoted to this 2D material, results from SEM, Raman and PL are considered to be well sufficient to give a reliable analysis of the material quality. Here is a non-exhaustive list of such articles.
Ranjuna, M. K., & Balakrishnan, J. (2023). "High temperature anomalous Raman and photoluminescence response of molybdenum disulfide with sulfur vacancies." Scientific Reports, 13(1), 16418.
Lin, Z., Liu, W., Tian, S., Zhu, K., Huang, Y., & Yang, Y. (2021). "Thermal expansion coefficient of few-layer MoS2 studied by temperature-dependent Raman spectroscopy." Scientific Reports, 11(1), 7037.
Chen, T., Zhou, Y., Sheng, Y., Wang, X., Zhou, S., & Warner, J. H. (2018). "Hydrogen-Assisted Growth of Large-Area Continuous Films of MoS2 on Monolayer Graphene." ACS Applied Materials and Interfaces, 10(8), 7304–7314.
Sojková, M., Siffalovic, P., Babchenko, O., Vanko, G., Dobročka, E., Hagara, J., Mrkyvkova, N., Majková, E., Ižák, T., Kromka, A., & Hulman, M. (2019). "Carbide-free one-zone sulfurization method grows thin MoS2 layers on polycrystalline CVD diamond." Scientific Reports, 9(1), 2001.
Yang, P., Yang, A.-G., Chen, L., Chen, J., Zhang, Y., Wang, H., Hu, L., Zhang, R.-J., Liu, R., Qu, X.-P., Qiu, Z.-J., & Cong, C. (2019). "Influence of seeding promoters on the properties of CVD grown monolayer molybdenum disulfide." Nano Research, 12(4), 823–827.
Moreover, we have already discussed in great depth strain effects in these same 2D material (grown with liquid precursors and in the same conditions) in Ref. 25 published last year. As this topic was exhaustively analyzed in that paper, adding results and discussion on strain effects in this paper would have resulted in a mere repetition.
Seravalli, L., et al. "Built-in tensile strain dependence on the lateral size of monolayer MoS2 synthesized by liquid precursor chemical vapor deposition." Nanoscale 15.35 (2023): 14669-14678.
- Although the paper confirms the influence of carrier gas flow rate in the CVD process, similar investigations have already been well-documented. This study does not provide significantly novel insights for readers.
It is true that there are numerous studies on optimizing growth conditions in the CVD process, but most of them (as pointed out in the introduction) deal with solid precursors and give results very different from the ones observed with liquid precursors. The novelty of this work lies primarily in the influence of the carrier gas flow when using liquid precursors, which represents an innovative approach compared to traditional methods. It should be noted that, no analysis on the effect of carrier flow rate when using liquid precursors has been reported so far. Also, we think that it is very relevant that the effect of carrier flow rate is rather different from the one occurring when using solid precursors.
Moreover, the achievement of large (more than 100 micron of lateral size) high-quality monolayers, such as those presented in the present study, is not a trivial result. We feel that our research contributes to a deeper understanding of the relationship between growth conditions and the final material quality, contributing to improve the quality of CVD-grown 2D MoS2.
Reviewer 3
This manuscript by Esposito et al. presents the growth of MoS2 nanomaterials using a hybrid synthesis approach, investigating the influence of the carrier gas. The authors chose a Mo-based liquid precursor for spin-coating, followed by sulfur treatment to grow MoS2 flakes. With increases in N2 flow rates, the size of MoS2 flakes increases, and the number of layers is modulated, resulting in excellent control over the formation of monolayer MoS2 flakes of a few micrometers in size. The MoS2 phase was confirmed using Raman spectroscopy, and the physical and optical properties were examined using scanning electron microscopy and photoluminescence spectroscopy. I would like to suggest addressing the comments below to improve the clarity and scientific rigor of this work. This work is interesting, and I hope that the revised manuscript can be accepted for publication in the journal Nanomaterials.
- The introduction section should suggest why MoS22D films are important to study by providing a variety of applications and fundamental optoelectronic processes.
We appreciate the reviewer's suggestion regarding the importance of MoS₂ 2D films and their applications and we have added some sentences to the introduction, providing some examples of applications in optoelectronics.
- The reasons for carrying out the growth process at 820°C and for choosing nitrogen gas as the carrier gas should be clarified in the experimental section.
We clarified the rationale behind selecting a growth temperature of 820°C and the choice of nitrogen as the carrier gas in the experimental section. The temperature was chosen based on data on the optimal growth window for MoS2 previously published by our group. Nitrogen is a very common carrier gas for the growth of MoS2, generally selected as it can prevent unwanted reactions during the synthesis process because of its inert nature. Furthermore its cost is lower than that of Argon. Such information has been added in the Experimental section.
Seravalli, L., et al. "Built-in tensile strain dependence on the lateral size of monolayer MoS2 synthesized by liquid precursor chemical vapor deposition." Nanoscale 15.35 (2023): 14669-14678.
- The modulation of lateral size according to increases in N2gas flow from 150 to 300 sccm is very interesting. The reason for choosing this flow rate range should be specified. I am curious why the authors limited their study to a flow rate of 300 sccm and what the pressure was during the processing. Did the N2 flow rate affect the pressure during the growth?
Our main objective is to achieve monolayers with satisfactory lateral sizes. While we have explored both higher and lower flow rates, we observed that reducing the flow rate leads to smaller flakes, as already seen at 150 sccm. Conversely, increasing the flow rates results in the formation of undesirable structures, which we aim to avoid. The process is at atmospheric pressure and the flow rate did not influence significantly the pressure during the growth.
We specified the reasons behind the range of flows considered in the revised manuscript in section 3 (red text).
- MoS2researchers are looking forward to growing continuous films of MoS2. While the increase in N2 flow rate results in larger MoS2 flakes, it would be beneficial to clarify whether it is possible to achieve a uniform continuous MoS2 film using the approach presented in this study.
We recognize the interest in growing continuous MoS₂ films. However, achieving a uniform continuous MoS₂ film via CVD following spin-coating of the liquid precursor presents significant challenges. In the literature, there are reports on positioning the substrates vertically instead of horizontally, which seems to facilitate this goal.
Wang, Shanshan, et al. (2016) "Substrate control for large area continuous films of monolayer MoS2 by atmospheric pressure chemical vapor deposition." Nanotechnology 27.8: 085604.
We have added a sentence on this point in the revised manuscript at the end of section 4.
- It is strongly recommended to add a table summarizing the progress of MoS2flake growth using various approaches to highlight the novelty of this study. Parameters such as the number of layers, flake size, precursor materials, processing temperature, substrate, and applications should be summarized.
We appreciate your recommendation to include a summary table comparing the progress of MoS2 flake growth using various approaches. We have added the table for sake of comparison and to highlight the novelty of our study. Parameters such as the number of layers, flake size, precursor materials, processing temperature, substrate, and applications were included in the new table. Due to the very large number of scientific articles on this material, this table cannot however be exhaustive by any means and it should be considered only as a guideline of standard values that can be found in the literature.
- It would be better to add a schematic of the synthesis procedure in Figure 1.
We agree that adding a schematic of the synthesis procedure would enhance the clarity of our manuscript. We have included Figure 1 (c) that visually represents the synthesis process, to help readers in understanding our hybrid approach.

Reviewer 2 Report
Comments and Suggestions for Authors
This paper investigates the influence of carrier gas flow rates on the synthesis of two-dimensional MoS₂ using liquid molybdenum precursors via chemical vapor deposition (CVD). By adjusting the flow rate of the carrier gas, the study analyzes variations in the morphology, number of layers, size, and other properties of MoS₂ flakes. It proposes a growth model that links sulfur provision with carrier gas flow rate and explains how this impacts the structure of MoS₂.
1. The paper primarily focuses on the impact of carrier gas flow rate on flake size and morphology but provides insufficient discussion of other important parameters such as temperature and pressure. These factors likely play a significant role in determining the quality and morphology of the flakes.
2. Although the sulfur supply model is proposed, the authors assume that sulfur availability is solely controlled by the carrier gas flow rate. It neglects the influence of other variables such as temperature gradients and gas-phase mixing, which may lead to inaccuracies in the model.
3. The study highlights the advantages of liquid molybdenum precursors over solid ones but fails to provide experimental comparisons between the two. Conducting experiments that directly compare different precursor types would strengthen the conclusions about their actual effects on MoS₂ growth.
4. While the paper includes data from Raman spectroscopy and SEM, the interpretation of these results is not in-depth. Specifically, the discussion on strain effects observed in Raman spectra is rather superficial, without a thorough exploration of their origins or mechanisms.
5. Although the paper confirms the influence of carrier gas flow rate in the CVD process, similar investigations have already been well-documented. This study does not provide significantly novel insights for readers.
Author Response

(The authors gave the same response as above.)

Reviewer 3 Report
Comments and Suggestions for Authors
This manuscript by Esposito et al. presents the growth of MoS2 nanomaterials using a hybrid synthesis approach, investigating the influence of the carrier gas. The authors chose a Mo-based liquid precursor for spin-coating, followed by sulfur treatment to grow MoS2 flakes. With increases in N2 flow rates, the size of MoS2 flakes increases, and the number of layers is modulated, resulting in excellent control over the formation of monolayer MoS2 flakes of a few micrometers in size. The MoS2 phase was confirmed using Raman spectroscopy, and the physical and optical properties were examined using scanning electron microscopy and photoluminescence spectroscopy. I would like to suggest addressing the comments below to improve the clarity and scientific rigor of this work. This work is interesting, and I hope that the revised manuscript can be accepted for publication in the journal Nanomaterials.
1. The introduction section should suggest why MoS2 2D films are important to study by providing a variety of applications and fundamental optoelectronic processes.
2. The reasons for carrying out the growth process at 820°C and for choosing nitrogen gas as the carrier gas should be clarified in the experimental section.
3. The modulation of lateral size according to increases in N2 gas flow from 150 to 300 sccm is very interesting. The reason for choosing this flow rate range should be specified. I am curious why the authors limited their study to a flow rate of 300 sccm and what the pressure was during the processing. Did the N2 flow rate affect the pressure during the growth?
4. MoS2 researchers are looking forward to growing continuous films of MoS2. While the increase in N2 flow rate results in larger MoS2 flakes, it would be beneficial to clarify whether it is possible to achieve a uniform continuous MoS2 film using the approach presented in this study.
5. It is strongly recommended to add a table summarizing the progress of MoS2 flake growth using various approaches to highlight the novelty of this study. Parameters such as the number of layers, flake size, precursor materials, processing temperature, substrate, and applications should be summarized.
6. It would be better to add a schematic of the synthesis procedure in Figure 1.

Author Response

(The authors gave the same response as above.)

Round 2
Reviewer 2 Report
Comments and Suggestions for Authors
Authors answered all questions. I think this manuscript can be accepted.
Reviewer 3 Report
Comments and Suggestions for Authors
The authors' revised manuscript shows notable improvements. The reporting of large-area MoS2 flakes is a significant contribution to the development of 2D-material-based optoelectronic devices. I recommend this study for publication and eagerly anticipate further contributions in the field of liquid-precursor grown 2D flakes.